# Tumor Treating Fields (TTFields) Concomitant with Sorafenib Inhibit Hepatocellular Carcinoma *In Vitro* and *In Vivo*

**DOI:** 10.3390/cancers14122959

**Published:** 2022-06-15

**Authors:** Shiri Davidi, Sara Jacobovitch, Anna Shteingauz, Antonia Martinez-Conde, Ori Braten, Catherine Tempel-Brami, Einav Zeevi, Roni Frechtel-Gerzi, Hila Ene, Eyal Dor-On, Tali Voloshin, Itai Tzchori, Adi Haber, Moshe Giladi, Adrian Kinzel, Uri Weinberg, Yoram Palti

**Affiliations:** 1Novocure Ltd., Haifa 3190500, Israel; sdavidi@novocure.com (S.D.); sgerstein@novocure.com (S.J.); ashteingauz@novocure.com (A.S.); amartinez@novocure.com (A.M.-C.); obraten@novocure.com (O.B.); cbrami@novocure.com (C.T.-B.); ezeevi@novocure.com (E.Z.); rfrechtel@novocure.com (R.F.-G.); hene@novocure.com (H.E.); edor-on@novocure.com (E.D.); tvoloshinsela@novocure.com (T.V.); itzchori@novocure.com (I.T.); ahaber@novocure.com (A.H.); weinberg@novocure.com (U.W.); yoram@novocure.com (Y.P.); 2Novocure GmbH, 81925 Munich, Germany; akinzel@novocure.com

**Keywords:** TTFields, Tumor Treating Fields, cancer treatment, hepatocellular carcinoma, autophagy, sorafenib, combination therapy

## Abstract

**Simple Summary:**

Hepatocellular carcinoma (HCC), an advanced liver cancer, has poor prognosis and limited treatment options. Tumor Treating Fields (TTFields) therapy is a novel antimitotic treatment, delivering electric fields that induce death and inhibit replication of cancer cells. We aimed to determine the effect of TTFields in HCC cells and an animal model, alone or in combination with sorafenib, an approved HCC treatment option. Human HCC cells were treated with TTFields at various frequencies to identify the most effective frequency. TTFields at 150 kHz were shown to induce anti-cancerous effects and to amplify such effects displayed by sorafenib. In animals, TTFields concomitant with sorafenib were more effective than either TTFields or sorafenib alone in reducing tumor volume, with the combination also leading to more cases of stable disease. Overall, this research demonstrates potential for concomitant TTFields and sorafenib application in the treatment of HCC.

**Abstract:**

Hepatocellular carcinoma (HCC), a highly aggressive liver cancer, is a leading cause of cancer-related death. Tumor Treating Fields (TTFields) are electric fields that exert antimitotic effects on cancerous cells. The aims of the current research were to test the efficacy of TTFields in HCC, explore the underlying mechanisms, and investigate the possible combination of TTFields with sorafenib, one of the few front-line treatments for patients with advanced HCC. HepG2 and Huh-7D12 human HCC cell lines were treated with TTFields at various frequencies to determine the optimal frequency eliciting maximal cell count reduction. Clonogenic, apoptotic effects, and autophagy induction were measured. The efficacy of TTFields alone and with concomitant sorafenib was tested in cell cultures and in an orthotopic N1S1 rat model. Tumor volume was examined at the beginning and following 5 days of treatment. At study cessation, tumors were weighed and examined by immunohistochemistry to assess autophagy and apoptosis. TTFields were found *in vitro* to exert maximal effect at 150 kHz, reducing cell count and colony formation, increasing apoptosis and autophagy, and augmenting the effects of sorafenib. In animals, TTFields concomitant with sorafenib reduced tumor weight and volume fold change, and increased cases of stable disease following treatment versus TTFields or sorafenib alone. While each treatment alone elevated levels of autophagy relative to control, TTFields concomitant with sorafenib induced a significant increase versus control in tumor ER stress and apoptosis levels, demonstrating increased stress under the multimodal treatment. Overall, TTFields treatment demonstrated efficacy and enhanced the effects of sorafenib for the treatment of HCC *in vitro* and *in vivo*, via a mechanism involving induction of autophagy.

## 1. Introduction

Liver cancer is the third leading cause of cancer death worldwide [1]. Hepatocellular carcinoma (HCC) accounts for 90% of primary liver cancer cases, is highly aggressive, and is associated with a high rate of recurrence and poor prognosis [2,3,4,5]. While HCC can potentially be treated curatively with ablation, surgical resection or liver transplantation at early disease stages, 85% of patients are diagnosed at an advanced stage and are only eligible for systemic therapy [2,5]. Chemotherapies, as single agents or combination treatments, have not shown a persuasive survival benefit, and due to their low tolerance in these hepatically challenged patients, are of limited use [3]. Sorafenib is a small molecule multikinase inhibitor which was the first systemic treatment approved for HCC. A phase II study demonstrated significant improvement in survival of patients with advanced HCC when treated with sorafenib relative to supportive care [6,7]. However, most HCC patients experience loss of sorafenib efficacy over time, apparently due to the development of resistance through the induction of autophagy to aid carcinoma cell survival under stress conditions [2,3,8,9]. As HCC mostly arises in immunosuppressive microenvironments [10], immunomodulators have been explored for treatment of this malignancy. Recently, the concomitant use of atezolizumab (anti-programmed death-ligand 1 (PD-L1) antibody) plus bevacizumab (anti-vascular endothelial growth factor (VEGF) antibody) was approved by the United States Food and Drug Administration (FDA) for first-line treatment of unresectable or metastatic HCC, based on the results of the IMbrave-150 study [11]. Combination of nivolumab (anti-programmed cell death protein 1 (PD-1) antibody) plus ipilimumab (anti-cytotoxic T lymphocyte-associated antigen-4 (CTLA-4) antibody) was approved for HCC patients previously treated with sorafenib, based on the CheckMate 040 trial [12].

Another avenue for the ongoing development of new treatment options for HCC include drugs/modalities that may be delivered together with sorafenib to increase efficacy and decrease resistance. One interesting possibility is concomitant therapy with Tumor Treating Fields (TTFields), a treatment modality that has already been demonstrated to augment the efficacy of sorafenib in preclinical models of glioblastoma (GBM) [13]. TTFields are low-intensity (1–3 V/cm RMS), intermediate-frequency (100–500 kHz), alternating electric fields with anti-mitotic effects on cancerous cells at cell-type specific frequencies. TTFields therapy is FDA approved for treatment of recurrent GBM, newly diagnosed GBM in combination with temozolomide, and unresectable malignant pleural mesothelioma (MPM) concomitant with pemetrexed and a platinum-based agent. TTFields are delivered loco-regionally and non-invasively via pairs of insulated ceramic arrays placed orthogonally on the patient’s skin overlaying the tumor site. The electric forces that comprise TTFields have been suggested to cause dielectrophoresis and dipole alignment, which disrupt mitotic spindle formation leading to aberrant mitosis and subsequent cell death in dividing cells [14,15,16]. Recently, TTFields have been shown to increase replication stress and impair DNA damage repair (DDR) mechanisms [17,18,19,20,21]. TTFields have also been shown to trigger autophagy in the progeny of cells dividing during TTFields application, in response to aberrant mitosis and genotoxic stress within daughter cells [22,23].

Autophagy is often considered as a ‘double-edged sword’ as it can be utilized as a survival strategy of cells, but when over-activated it can mediate cell death [9]. Autophagy has been suggested to be anti-tumorigenic in normal liver by suppressing tumor initiation, but also to support carcinoma cell survival in response to chemotherapy [2]. We hypothesized that concomitant application of sorafenib and TTFields may increase stress levels enough to tilt autophagy towards the cell death pathway. Accordingly, we examined the effect of TTFields in HCC cellular and animal models, alone and in combination with sorafenib.

## 2. Materials and Methods

### 2.1. Materials

Media, supplements and phosphate buffered saline (PBS) were purchased from Biological Industries Ltd. (Beit Haemek, Israel). Ketamine, xylazine, and isoflurane were acquired from Vetmarket Ltd. (Shoham, Israel) Matrigel was obtained from Bactlab Diagnostic Ltd. (Caesarea, Israel). Paraformaldehyde (PFA), Triton, 4′,6-diamidino-2-phenylindole (DAPI), and chloroquine diphosphate (CQ) were purchased from Sigma-Aldrich (Rehovot, Israel).

### 2.2. Cell Cultures

HepG2 cells (ATCC) were grown in Eagle’s Minimum Essential Medium (EMEM) supplemented with 10% (*v/v*) fetal bovine serum (FBS) and penicillin-streptomycin (50 µg/mL). Huh-7D12 (Sigma-Aldrich, Rehovot, Israel) and N1S1 (ATCC) cells were grown in Dulbecco’s Modified Eagle’s Medium (DMEM), supplemented with 10% (*v/v*) FBS, 2 mM L-glutamine and penicillin-streptomycin (50 µg/mL). All cells were grown in a 37 °C humidified incubator supplied with 5% CO_2_.

### 2.3. In Vitro Experiments

HepG2 and Huh-7D12 cell suspensions (500 µL, 25 × 10^3^ cells/plate) were placed as a drop in the center of 35-mm inovitro™ dishes composed of high dielectric constant ceramic (lead magnesium niobate–lead titanate (PMN-PT)), with two perpendicular pairs of transducer arrays printed on their outer walls. Cells were incubated overnight at 37 °C to allow attachment to the dish, and then 2 mL of fresh media was added. TTFields were then applied to the cells using the inovitro system [14,15,16]. Frequency scans (100–400 kHz) were conducted for 72 h with TTFields applied at intensities of 1.0 V/cm RMS for HepG2 and 1.7 V/cm RMS for Huh-7D12, followed by cell count measurements. Frequency scans were also performed in the N1S1 murine cell line.

All subsequent tests were performed with HepG2 and Huh-7D12 cells at 150 kHz TTFields. Cells were analyzed to determine cell count, apoptosis, and granularity, and surviving cells were further evaluated for their clonogenic potential. To test for the induction of autophagy, TTFields were applied for 24 or 48 h, with or without 20 μM CQ incubated during the last 3 h of treatment, followed by immunofluorescence and immunoblotting examinations. TTFields-sorafenib combinations were evaluated by co-application of TTFields with 0.1–3 µM sorafenib tosylate (BioVision, Milpitas, CA, USA) followed by cell count, colony, and apoptosis assays. Control, TTFields, 3 µM sorafenib, and TTFields + 3 µM sorafenib treated cells, that were incubated with 20 μM CQ during the last 3 h of treatment, were further analyzed by immunoblotting.

### 2.4. Flow Cytometry Analysis of Cell Count, Granularity and Apoptosis

Cells were counted using iCyt EC800 (Sony Biotechnology, San Jose, CA, USA) flow cytometer, and expressed as a percentage relative to the control. Granularity was determined from the median side scatter values of the flow cytometry readings and expressed as a percentage relative to control. Apoptosis was determined by double-staining of the cells with FITC-conjugated Annexin V (AnnV) and 7-Aminoactinomycin D (7-AAD) using a commercially available kit (BioLegend, San Diego, CA, USA) as per manufacturer’s instructions, and data acquisition at 525/50 nm and 665/30 nm, respectively. The data were analyzed using the iCyt EC800 software.

### 2.5. Colony Forming Assay and Overall Effect Calculation

At the end of treatment, cells were harvested, re-plated into 6-well tissue culture plates (300 cells/well), and grown for an additional 7–14 days in chemotherapy-free media. Colonies were stained with 0.5% crystal violet solution and counted. Clonogenic effect was calculated as percentage relative to control, and the overall treatment effect was defined as the product of cell count and the corresponding clonogenicity.

### 2.6. LC3 Foci Detection by Fluorescent Microscopy, In Vitro

Cells were fixed with ice-cold methanol for 10 min, serum-blocked and stained with anti-microtubule-associated protein 1 LC3B primary antibody (rabbit polyclonal, Novus Biologicals, Littleton, CO, Canada; NB600-1384, 1:200) followed by Alexa Fluor 488-conjugated secondary antibody (Jackson ImmunoResearch, Cambridge, UK; 711-545-152, 1:300) and DAPI (Sigma-Aldrich, Rehovot, Israel; 32670, 1:1000) for nuclei counterstaining. Images were collected using an LSM 700 laser scanning confocal system (Zeiss, Gottingen, Germany), attached to an upright motorized microscope (ZeissAxio Imager Z2, Gottingen, Germany). Mean number of LC3 foci per cell was determined using ImageJ software with a median filter to find the local maxima, and expressed as a percentage relative to control.

### 2.7. Western Blot Analysis

Cell extracts were prepared and subjected to Western blot analysis (30 μg protein) as previously described [14], using anti-LC3B (rabbit polyclonal, Novus NB600-1384, 1:1000), anti-beclin-1 (mouse monoclonal, Santa Cruz Biotechnology Inc, Dallas, TX, USA; 48341, 1:500), anti-GRP78 (mouse mono-clonal, Santa Cruz Biotechnology Inc, Dallas, TX, USA; 376768, 1:500), and anti-cleaved-Poly (ADP-ribose) polymerase (PARP) (rabbit monoclonal, cell signaling, 94885, 1:1000) primary antibodies, followed by incubation with horseradish peroxidase (HRP)-conjugated secondary antibody (Abcam, Cambridge, UK; goat anti-mouse ab97023 or goat anti-rabbit ab6721, 1:1000). A chemiluminescent substrate (Immobilon Forte, Millipore, Burlington, MA, USA) was used for visualization, and signals were recorded on GeneGnome XRQ gel imager (AlphaMetrix Biotech, Rödermark, Germany). Densitometric readings were quantified with FIJI software for LC3BII/I ratio and expressed as fold change relative to control.

### 2.8. In Vivo Experiment

Animal housing and anesthesia as well as array composition and placement procedure were previously described [24]. Male SD rats (age: 7–8 weeks; weight: 150–200 g; Envigo Ltd., Jerusalem, Israel) were inoculated with N1-S1 cells (50,000 suspension in 10 µL serum free DMEM medium, diluted 1:1 in Matrigel) into the left hepatic lobe. Rats were allowed to recuperate and develop tumors for 1 week before treatment initiation. Tumor volume and location were examined by magnetic resonance imaging (MRI), and only rats with a verified tumor at the correct location and with a volume between 30–100 mm^3^ were included in the experiment. Qualified animals (*n* = 52) were randomized into four study groups: (1) sham (heat), vehicle (saline); (2) TTFields, vehicle; (3) sham, sorafenib (Santa Cruz); and (4) TTFields, sorafenib. Throughout the study the rats were housed in individual cages to prevent tangling of the wires connected to the device. TTFields (2.4 V/cm RMS; 150 kHz; Novo-TTF100L device) were applied continuously for 5 days through two pairs of perpendicular arrays placed around the tumor, each pair applying unidirectional TTFields for 1 s intermittently [21,24]. Device usage data were recorded to ensure animals successfully received therapy for ≥18 h/day, as per clinical recommendations for maximizing treatment benefits [25,26]. Sham arrays were identical in size and shape to the TTFields arrays, generated equivalent heat (38.5 °C), and were placed on the torso of the animals at the same orientation as the treatment arrays. Sorafenib (10 mg/kg/day) or vehicle were intraperitoneally injected each day of the treatment period (total of five times). Tumor volume was examined by MRI after array disconnection, and fold change relative to the start point was calculated. A day later, the animals were sacrificed, and liver tumors removed and weighed.

### 2.9. MRI Tumor Volume Assessment

MRI scans were acquired using the ICON (1 Tesla, desktop) MRI scanner (Bruker Biospin, Ettlingen). Animals were placed in a dedicated rat body coil in a prone position, and a T_2_ weighted coronal anatomical image was scanned with a RARE sequence using the following parameters: excitation time, 51 ms; repetition time, 1900 ms; average number of scans, 8; number of slices, 10; slice thickness, 1 mm; matrix size, 140 × 140; FOV, 55–65 mm; in plane resolution −0.36 × 0.36 mm; and acquisition time, 4 min 18 s. The tumors were manually segmented using the ITK-SNAP software version 3.6.0-rc1 free, which calculated tumor volume as the sum of all the segmented areas.

### 2.10. Immunohistochemistry Analysis

Immunohistochemical staining was performed on formalin-fixed and paraffin-embedded 4-µm-thick tumor sections using the Leica BOND-MAX system (Leica Biosystems Ltd., Newcastle, UK). Tissues were pretreated with BOND epitope-retrieval solution for 20 min, followed by 30 min incubation with anti-beclin-1 (mouse-anti-rodent, Santa Cruz 48341, 1:100), anti-GRP78 (rabbit-anti-rodent, Abcam 32618, 1:100), or anti-cleaved-PARP (rabbit-anti-rodent, Cell Signaling 94885, 1:100) primary antibodies. The Leica Refine-HRP kit (Leica Biosystems DS9800) was used for detection (goat-anti-rabbit, HRP conjugated secondary antibody therein) and to counter-stain with hematoxylin. The slide was scanned, and the CaseViewer software was used to exclude non-tumor areas. The signals of the stained protein and the nuclei were resolved by color deconvolution and quantified separately using ImageJ software. Average signal per cell or percent of positive cells was calculated.

### 2.11. LC3 Foci Detection by Fluorescent Microscopy, In Vivo

Paraffin-embedded tumor sections were deparaffinized with HistoChoice (Sigma-Aldrich, H2779) and rehydrated with graded alcohol treatments. Antigen retrieval was carried out by microwave treatment for 22 min in citrate buffer (pH 6.0). Sections were blocked in 10% normal donkey serum in PBS and incubated overnight with primary antibody LC3B (rabbit polyclonal, Novus NB600-1384, 1:100) followed by secondary antibody (Alexa Flour 488, Jackson ImmunoResearch 711-545-152, 1:300) and DAPI incubation for nuclei visualization. Average green intensity per image was calculated with ImageJ software, based on three different areas of similar size chosen in a blinded manner.

### 2.12. Statistical Analysis

All *in vitro* experiments were repeated at least three times, and data are presented as mean ± standard error of the mean (SEM). For *in vivo* studies, data are presented as mean ± standard deviation (SD). Statistical significance was calculated using GraphPad Prism 8 software (La Jolla, San Diego, CA, USA), with the specific tests used mentioned in figure legends. Differences were considered significant at values of: * *p* < 0.05, ** *p* < 0.01, *** *p* < 0.001, and **** *p* < 0.0001.

## 3. Results

### 3.1. Efficacy of TTFields in HCC Cells

Treatment of HepG2 and Huh-7D12 cells with TTFields for 72 h across a frequency range of 100–400 kHz revealed maximal effect at 150 kHz, with cell counts declining to 40.0 ± 3.8% and 36.5 ± 3.5% relative to control, respectively (Figure 1a). All experiments were conducted with TTFields applied at intensities of 1.0 V/cm RMS for HepG2 cells and at 1.7 V/cm RMS for Huh-7D12 cells to produce an effect of the same magnitude in both cell lines. The colony-forming ability of the cells surviving treatment decreased significantly, to 36.8 ± 8.0% of control for HepG2 cells and to 32.0 ± 18.0% relative to control for Huh-7D12 cells (Figure 1b). The resulting overall effect of TTFields application was 20.1 ± 6.7% and 19.7 ± 10.2% for HepG2 and Huh-7D12 cells, respectively (Figure 1c). Apoptosis analysis showed a reduction of the live cells fraction (AnnV-7ADD-stained cells) from 87.5 ± 2.3% for non-treated HepG2 cells to 72.6 ± 4.3% following delivery of TTFields, and from 76.6 ± 4.4% for non-treated Huh-7D12 cells to 63.2 ± 6.1% following application of TTFields (Figure 1d). However, this effect was only significant in HepG2 cells.

### 3.2. TTFields Elevate Autophagy in HCC Cell Lines

HepG2 and Huh-7D12 cells treated with 150 kHz TTFields for 72 h exhibited significant elevations of 20% and 9%, respectively, in side-scatter parameters, i.e., cellular granularity (Figure 2a). To explore whether this phenomenon is indicative of increased autophagy, fluorescent microscopy was employed to detect microtubule-associated protein light chain 3 (LC3). Higher levels of LC3 foci were seen in cells treated with TTFields relative to control cells (Figure 2b). Experiments performed with the addition of chloroquine (CQ), an inhibitor of lysosome degradation allowing for measuring autophagic flux, showed higher puncta levels than without CQ, indicating that the observed effect was upregulation of the autophagy process rather than reduced autophagosome turnover. Quantification of immunofluorescent staining (Figure 2c) and immunoblots (Figure 2c) of experiments performed with CQ added during the last 3 h of treatment revealed elevation of LC3 foci formation and an increased LC3-II to LC3-I ratio, respectively, at both 24 and 48 h of TTFields application in both cell lines. However, autophagy kinetics seems to be faster in the HepG2 cells, in which LC3 markers are lower at 48 versus 24 h, whereas elevation is seen from 24 to 48 h for the Huh-7D12 cells.

### 3.3. Combination of TTFields with Sorafenib Enhances Treatment Efficacy In Vitro

HepG2 and Huh-7D12 cells treated for 72 h with sorafenib displayed dose-response relationships for cell count (Figure 3a), clonogenic (not shown), and overall effects (Figure 3b), at a concentration range of 0.1–3 µM. Combining sorafenib with TTFields had additive effects in both cell lines. Cellular apoptosis was also induced by sorafenib in a dose-dependent manner for both HepG2 and Huh-7D12 cells (Figure 3c). Co-application of sorafenib and TTFields significantly decreased the live cells fraction (AnnV-7ADD-stained cells) relative to sorafenib alone in HepG2 cells, but not in Huh-7D12 cells.

### 3.4. Autophagy–Apoptosis Interplay For Treatment with Concomitant TTFields and Sorafenib

In order to investigate the mechanism of action of TTFields-sorafenib co-application, HepG2 and Huh-7D12 cells were treated for 6, 24, or 48 h with TTFields, sorafenib (3 µM), or the two modalities together, and then examined for expression levels of various proteins. For HepG2 cells, the autophagy marker beclin-1 demonstrated elevation after 6 h of treatment, which was later replaced with diminished expression levels (Figure 3d). This type of behavior was seen in all treatment groups, but was most pronounced for TTFields-sorafenib co-application. The autophagy marker LC3-II/LC3-I also displayed such bi-phasic characteristics, but with a somewhat slower kinetics, showing some elevation at 6 h of treatment, but higher elevation at the 24 h time point (Figure 3d). As in the case of beclin-1, the magnitude of the effect was higher for co-treatment of TTFields and sorafenib relative to the monotherapies. GRP78, a marker of ER stress, remained low in all treatment groups for 6 and 24 h of treatment, but demonstrated elevated levels at the later, 48-h time point (Figure 3e). The apoptosis marker cleaved PARP displayed increased expression in the combined group already after 24 h, elevating even further after 48 h of treatment. For the monotherapies, cleaved PARP increase was only evident at 48 h of treatment, and to a lower extent than that in the co-treatment group (Figure 3f). The slower kinetics of the autophagy–apoptosis path in the Huh-7D12 cells, as seen from the elevation of LC3 after as much as 48 h (Figure 2c,d), prevented from detecting such changes in the levels of these markers in this cell line (Appendix A).

### 3.5. Concomitant TTFields with Sorafenib Enhances Treatment Efficacy In Vivo

The efficacy of combining TTFields with sorafenib relative to each modality alone was examined in the N1S1 HCC rat orthotopic model (timeline in Figure 4a). *In vitro* experiments confirmed that the 150 kHz TTFields frequency found to be optimal for treatment of the human cell lines was also optimal for treatment of the murine N1S1 cells used for the *in vivo* study (Appendix A). It is also worth mentioning that the N1S1 murine cells used for this study, like the HepG2 cells, are p53 wild type. The intensity of TTFields applied to the rat liver was in good agreement with values obtained in field intensity simulations in humans [27], and all rats reached the required usage limit of ≥18 h/day. During the treatment period, average tumor volumes of control animals (*n* = 11) increased 5.9-fold (Figure 4b,c and Appendix A for tumor images). For animals treated with TTFields (*n* = 15) or sorafenib (*n* = 10), tumor growth was significantly lower, 3.3- and 2.3-fold, respectively. In the TTFields-sorafenib combination group (*n* = 16), a 1.6-fold increase in tumor volume within the treatment period was observed, a growth significantly lower than that for control or for each treatment alone. Furthermore, in the control and sorafenib groups all animals displayed tumor growth during the experiment, while in the TTFields and TTFields-sorafenib groups, 20% and 44% of the animals, respectively, exhibited stable disease with no enlargement of the tumor. Average tumor weights at study cessation were 408 ± 338 mg for control, 229 ± 236 mg for TTFields, 153 ± 93 mg for sorafenib, and 104 ± 75 mg for the TTFields-sorafenib combination (Figure 4d). Tumor histology and immunostaining for beclin-1 and LC3, GRP78, and cleaved PARP were performed to examine autophagy, ER stress, and apoptosis levels, respectively. Beclin-1 levels were increased more than 4-fold relative to control in all treatment groups, while intensity of LC3 staining was increased about 3-fold relative to control in the individual TTFields and sorafenib groups, but only 2-fold in the combination group (Figure 4e). GRP78 levels in the groups treated with TTFields or sorafenib alone remained unchanged from the control, but were elevated 2-fold in the TTFields plus sorafenib group (Figure 4f). Additionally, the percentage of cells positive for cleaved PARP was significantly higher relative to control only in the combination group (Figure 4g).

## 4. Discussion

The current research aimed to examine the preclinical efficacy of TTFields for treating HCC, when applied as monotherapy or concomitant with sorafenib, with the latter being an approved first-line treatment for HCC. It has been previously shown that maximal effectivity of TTFields occurs at a different frequency for different cancer types, owing to the specific electrical properties of the cells [15,16]. Hence, the first step in applying TTFields to a new tumor type includes frequency scans. The effect of TTFields on reducing cell count in both HepG2 and Huh-7D12 human HCC cell lines was maximal at a frequency of 150 kHz. Delivery of TTFields at 150 kHz for 72 h also significantly reduced the colony forming ability of the cells, indicating that daughter cells formed under TTFields were damaged to a degree that prevented their continued proliferation. Previously, it has been shown that TTFields induce chromosomal aberrations, aneuploidy and multinucleation in progeny formed under application of TTFields, which may preclude further replication and support this long-term response to TTFields [14]. The lower TTFields intensity required for treatment of HepG2 relative to Huh-7D12 cells for achieving the same level of efficacy suggests higher sensitivity of the former to TTFields. Application of TTFields to HCC cells also induced cellular apoptosis, with a greater cytotoxic effect seen in HepG2 relative to Huh-7D12 cells. A reverse correlation has previously been shown between efficacy of TTFields and cellular doubling time [14]. However, this was not the case in the current study, as the doubling time of Huh-7D12 cells is shorter. Thus, this indicates that factors other than replication rate are involved in facilitating the effects of TTFields in these HCC cells. One such factor may be p53 status, which is wild-type in HepG2 and mutated in Huh-7D12. Indeed, previous work demonstrated lower TTFields-induced apoptosis in cell lines with mutated p53 [28].

Increased cellular granularity, amplified lipidation of LC3 (from LC3-I to LC3-II), and elevated levels of LC3 foci indicated that cells exposed to TTFields were undergoing increased autophagy, similar to what has been reported in GBM and non-small cell lung carcinoma (NSCLC) cells [22,23]. CQ is an inhibitor of lysosome degradation, commonly used to decipher whether the elevation of LC3 is due to upregulation of the autophagy process or reduced autophagosome turnover [29]. The higher TTFields-induced elevation of LC3 seen in the presence of CQ suggests that the observed phenomenon is due to increased autophagic flux, rather than decreased autolysosome degradation. While autophagy serves as a survival strategy of cells, when stress levels continue raising it may be over-activated and mediate cell death [9]. The faster autophagy kinetics seen for the HepG2 relative to Huh-7D12 cells following application of TTFields is in agreement with the higher apoptosis levels displayed by this cell line, and may serve as an additional rational for the higher efficacy of TTFields against it. Examination of the reasons for faster autophagy in HepG2 relative to Huh-7D12 cells is out of the scope of this work.

The effect of TTFields when co-applied with sorafenib was examined in cell cultures and animals. TTFields augmented the efficacy of sorafenib in HepG2 and HuH-7D12 cells lines, in regard to cell count, colony formation, and overall effect. In the case of HepG2 cells, concomitant application of TTFields and sorafenib led to elevated apoptosis. Kinetic examination in the HepG2 cells revealed elevation in autophagy levels as early as 6 h of TTFields or sorafenib treatment, which diminished and were replaced with ER stress and apoptosis for 48 h of treatment. These results are in line with a previous study that focused on the effects of sorafenib on such markers in HepG2 cells [30]. The higher changes in expression levels and faster kinetics when TTFields and sorafenib were applied together rather than alone indicate higher stress levels imposed on the cells in the former case.

In the HCC animal model, the acute effects of TTFields and sorafenib were examined. Due to the large tumors developed in the control group and the stress experienced by the animals as a result of the individual housing and motility limitations imposed by the sham and TTFields arrays, longer treatment durations were not feasible. In line with the *in vitro* results, TTFields applied at 150 kHz, displayed efficacy *in vivo*, and also added to the anti-tumor effects of sorafenib, with lower tumor volume fold change, lower tumor weight, and a higher number of stable disease cases for the combined treatment relative to the monotherapies. Since the optimal TTFields frequency has been shown to be dependent on the electrical properties of the cells and it is not clear how much effect the tumor microenvironment has on these properties, and because it is technically problematic to perform TTFields frequency scans *in vivo*, the frequency detected in the cell cultures was also employed for the animal studies. Interestingly, while TTFields and sorafenib each significantly increased autophagy within the tumor relative to control, elevation by the joint treatment was lower. On the other hand, the combination group displayed a significantly higher level of ER stress and apoptosis within the tumor relative to the control group, whereas almost no elevation was seen in the monotherapy groups. The lower autophagy accompanied by the higher ER stress and apoptosis displayed in the conjunction group relative to the monotherapies groups following 6 days of treatment suggests that these animals were pushed further along the autophagy–apoptosis kinetic timeline due to the higher levels of stress experienced by these animals, in accordance with the results described for the cell cultures. Overall, these results suggest that while tumor cells upregulate autophagy as a defense mechanism against TTFields and sorafenib alone, when the two treatments were combined the increased stress tilted the scale towards cellular death. Additional research is needed to address specific signaling pathways involved in this induction of autophagy by TTFields.

With the introduction of immunotherapies as treatment strategies for HCC, the landscape for treating this malignancy has been evolving. TTFields have recently been shown in preclinical models to induce antitumor immunity and have demonstrated benefit in combination with anti-PD-1 therapy [23], indicating broad applicability of TTFields therapy and promise regarding the concomitant treatment approach in the emerging era of immuno-oncology. In line with that, the induction of autophagy is a key driver of immunogenic cell death [31], thus warranting additional examination of immunotherapies and TTFields therapy in HCC.

## 5. Conclusions

TTFields were identified to be most efficient for treatment of HCC cells at 150 kHz, and this frequency further demonstrated *in vivo* efficacy. Induction of autophagy by TTFields shown here in HCC was previously demonstrated in GBM [13,22,32,33] and Lewis lung carcinoma [23], indicating this is a common outcome of TTFields. The increased efficacy displayed by the concomitant use of TTFields with sorafenib suggests that other combinations of TTFields with autophagy-inducing agents, such as rapamycin [34], should be pursued. As autophagy induction is also involved in immunogenic cell death [31], examinations of TTFields concomitant with immunotherapies is also warranted. Results of the phase II HEPANOVA study (NCT03606590) examining TTFields therapy (150 kHz) plus sorafenib for patients with advanced HCC demonstrated the safety and preliminary efficacy of this combination [35].

## Figures and Tables

**Figure 1 cancers-14-02959-f001:**
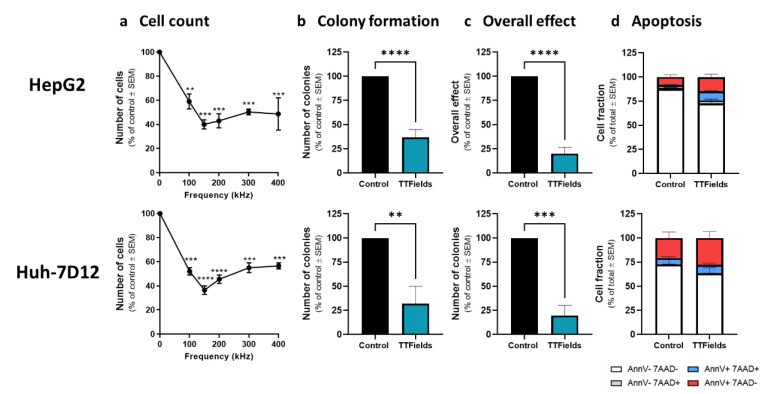
Efficacy of TTFields for treatment of HCC cells. HepG2 and Huh-7D12 cells were treated with TTFields (1.0 and 1.7 V/cm RMS, respectively) for 72 h. Cell counts were determined following treatment with TTFields at a frequency range of 100–400 kHz (**a**). Values are mean (*N* ≥ 3) ± SEM. ** *p* < 0.01, *** *p* < 0.001, and **** *p* < 0.0001 relative to control; one-way ANOVA. Clonogenicity (**b**), overall effect (**c**), and apoptosis (**d**) were examined for treatment of the cells with 150 kHz TTFields. Values are mean ± SEM. ** *p* < 0.01, *** *p* < 0.001, and **** *p* < 0.0001 relative to control; For apoptosis assay live cells: *p* < 0.05 for HepG2 and *p* = 0.14 for Huh-7D12 relative to control; Student’s T-test. ANOVA = analysis of variance; HCC = hepatocellular carcinoma; RMS = root mean square; SEM = standard error of the mean; TTFields = Tumor Treating Fields.

**Figure 2 cancers-14-02959-f002:**
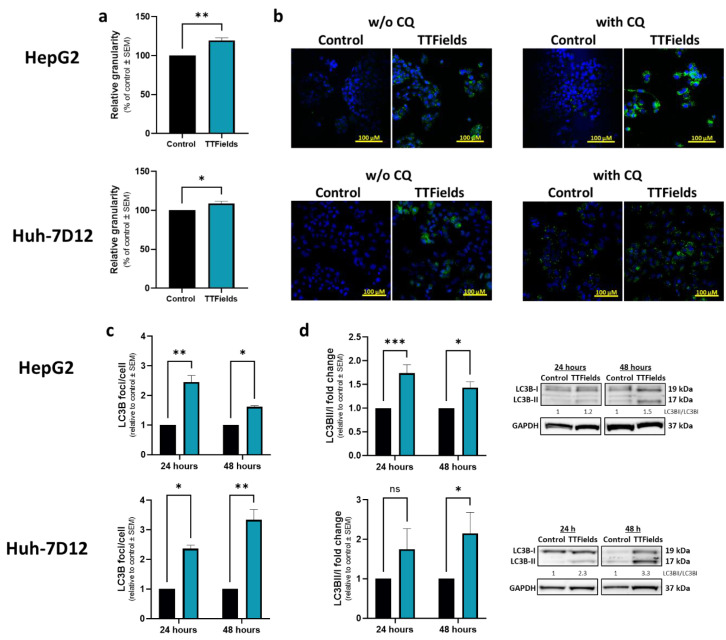
Effects of TTFields on autophagy levels in HCC cells. HepG2 and Huh-7D12 cells were treated with 150 kHz TTFields (1.0 and 1.7 V/cm RMS, respectively) for 72 h, and median side-scatter values were measured as representing changes in cellular granularity (**a**). Cells treated for 48 h were examined for LC3 foci formation by fluorescent microscopy, with or without addition of CQ during the last 3 h of the experiment (**b**), LC3 in green staining and DAPI in blue. Experiments performed with 24 or 48 h application of TTFields with addition of CQ during the 3 final hours of the experiments were quantified for LC3 foci formation by immunofluorescence (**c**), and for LC3-II to LC3-I ratio by immunoblotting (**d**). Values are mean (*N* ≥ 3) ± SEM. * *p* < 0.05, ** *p* < 0.01, and *** *p* < 0.001, relative to control; two-way ANOVA. ANOVA = analysis of variance; CQ = chloroquine diphosphate; DAPI = 4′,6-diamidino-2-phenylindole; HCC = hepatocellular carcinoma; ns = non-significant; RMS = root mean square; SEM = standard error of the mean; TTFields = Tumor Treating Fields.

**Figure 3 cancers-14-02959-f003:**
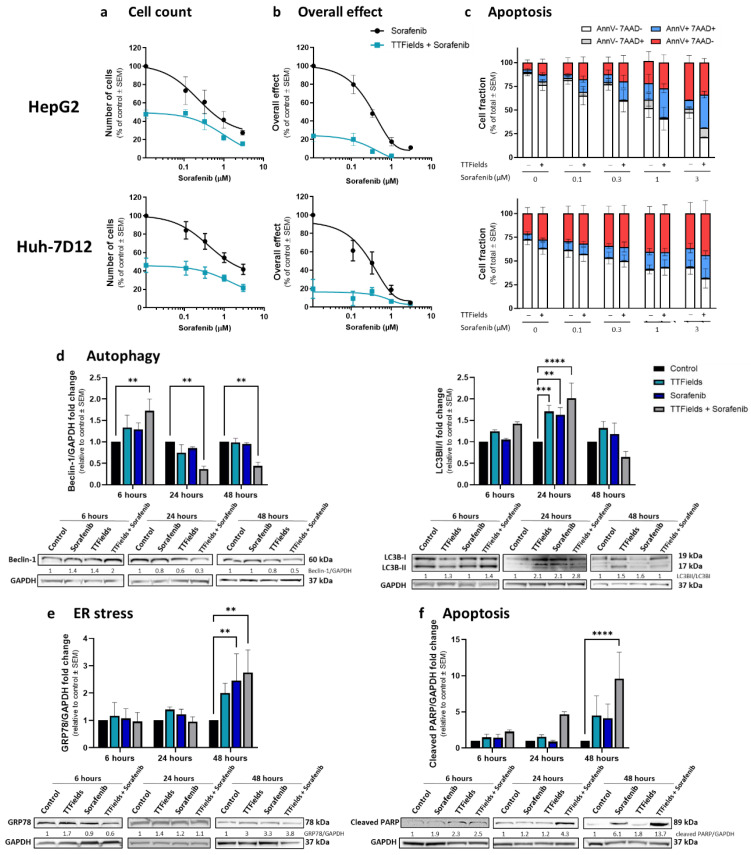
*In vitro* efficacy and mechanism of action of TTFields in combination with sorafenib. HepG2 and Huh-7D12 cells were treated for 72 h with various concentrations of sorafenib, alone or in combination with 150 kHz TTFields, followed by examination of cell count (**a**), overall effect (**b**), and apoptosis (**c**). Values are mean (*N* ≥ 3) ± SEM. For the dose effect of sorafenib: *p* < 0.001 for cell count and apoptosis, and *p* < 0.005 for overall effect. For the effect of TTFields versus sorafenib alone: *p* < 0.001 for cell count and overall effect, *p* < 0.01 and *p* = 0.16 for apoptosis assay live cells in HepG2 and Huh-7D12 cells, respectively; two-way ANOVA. HepG2 cells were treated for 6, 24, or 48 h with 150 kHz TTFields, 3 µM sorafenib, or the two treatments combined, followed by Western blot examination of the autophagy markers beclin-1 and LC3 (**d**), the ER stress marker GRP78 (**e**), and the apoptosis marker cleaved PARP (**f**). Values are mean (*N* ≥ 3) ± SEM. ** *p* < 0.01, *** *p* < 0.001, and **** *p* < 0.0001 relative to time-respective control; two-way ANOVA. ANOVA = analysis of variance; SEM = standard error of the mean; TTFields = Tumor Treating Fields.

**Figure 4 cancers-14-02959-f004:**
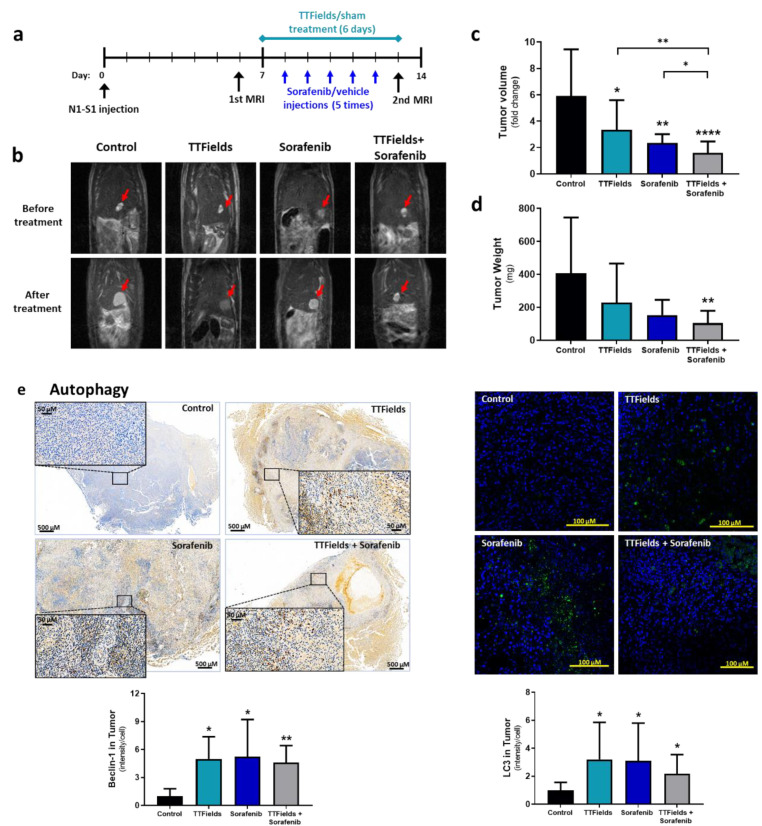
*In vivo* efficacy of TTFields in combination with sorafenib. Rats (*n* = 52) were inoculated orthotopically with rat HCC N1S1 cells, and treated with sham-vehicle (*n* = 11), TTFields alone (*n* = 15), sorafenib alone (*n* = 10), or TTFields + sorafenib (*n* = 16), according to the depicted timeline (**a**). MRI images were captured at the start and end of the study ((**b**), representative images), from which tumor volume fold change was calculated (**c**). At study end, tumors were removed and weighed (**d**), and tumor slices were subjected to immunohistochemical analysis for beclin-1 and LC3 (**e**), GRP78 (**f**), and cleaved PARP (**g**). Values are mean ± SD. * *p* < 0.05, ** *p* < 0.01, *** *p* < 0.001, and **** *p* < 0.0001 relative to control for labels above bars, or between indicated groups; Student’s T-test. HCC = hepatocellular carcinoma; MRI = magnetic resonance imaging; SD = standard deviation; TTFields = Tumor Treating Fields.

## Data Availability

All data generated or analyzed during this study are included in this article. Further inquiries can be directed to the corresponding author.

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
