# Peer review of "Tumor Treating Fields (TTFields) Concomitant with Sorafenib Inhibit Hepatocellular Carcinoma In Vitro and In Vivo"

_cancers, 2022, doi:10.3390/cancers14122959_

Round 1

Reviewer 1 Report

Cancers-1602413, Tumor Treating Fields Concomitant with Sorafenib Inhibit Hepatocellular Carcinoma in Vitro and in Vivo.

In this manuscript, the authors, Davidi et al have investigated the effect of TTFields in HCC cells and an animal model, alone or in combination with sorafenib. Overall, the authors believe that this research demonstrates potential for concomitant TTFields and sorafenib application in the treatment of HCC.

I read the manuscript with interest and commend the authors for the work done in the area of liver cancer treatment. However, I feel that the experiments around this concept were not designed well enough to support their new findings. I have raised the following concerns which is necessary to make this manuscript more scientifically interesting.

  1. What is the rationale of using 150 kHz TTFields? TTFields at 100 kHz has also shown significant differences as cytotoxicity at p<0.01 and p<0.001 in HepG2 and Huh-7D12 cells which have around 60% and 50% of cell survival. Why did the authors use high frequency of TTFields though the cytotoxicity was also observed in lower dose (Fig 1A).
  2. What is the ‘n’ number of the samples and experiments in each group? Also mention the ‘n’ number in all the experiments involved for invitro and invivo.
  3. LC3 is increased in all the groups except in control tissue, however, cell death was increased only in combined group. How can the authors correlate autophagy with apoptosis? It is not so trustworthy that the only expression of LC3B indicate the treated condition have increased autophagy in the tumor tissue. Other important autophagy and degradation markers like Beclin1 and P62 need to be shown to reflect the regulatory mechanism of TTFields, as well as for the combined treatment with Sorafenib.
  4. What was the total weight of livers in all groups? Please update the images of whole liver showing tumors on it, or at least pictures of tumors which were removed after sacrifice, if applicable.

Several data have shown very high error bars in each group (especially Fig Can the author provide the tumor images which were harvested from mice?

  1. In the IHC experiment for PARP, the authors have incubated with primary antibody only for 30 min? Is that timing enough to get protein expression? What source of secondary antibody was used? In addition to IHC, I suggest performing western blot using PARP antibody where the full length and cleaved bands are observed in the same blot.
  2. The figure number is mislabeled in Fig 4D-LC3 Immunofluorescence.
  3. Why there is no error bar in the control group of all bar graphs? Please include error bars and re-calculate the statistical analysis for all the data wherever missing.
  4. Figure legends of Fig2D is missing. Fig C is repeated in the legend. Please label the figures appropriately. It is so frustrating to understand.
  5. There is no data in Fig 4A. Whole data is missing but have explained in the result section and in figure legends.??
  6. Line 337 seems over statement since the data are not shown in the manuscript.
  7. Typos: Line 111, ‘invitro’ spelling is not correct. Double ‘and’ in Line 342. Need language and grammar check.

Reviewer 2 Report

This study identified the efficacy of TTFields treatment for hepatocellular carcinoma and the combined effect of sorafenib in vitro and in vivo

The authors suggested that the therapeutic effects of the combination were apoptosis via autophagy. 

   This is an interesting scientific study, as it concerns the issue of hepatocellular carcinoma, it also has a clinical aspect. 

   The authors indicated that TTField had potential to be a new treatment option of hepatocellular carcinoma. 

      The materials and methods section are elaborated in the details. 

         The present study was examined in terms of autophagy and apoptosis in the antitumor effects of TTField and sorafenib. 

        However, the exact mechanism of the combination therapy-induced cell death is not yet known as the activation of autophagy in the combination therapy was not increased. 

         Therefore, I expect further additional examination in the future.

      References contain mostly publications printed in the last 10 years. 

Reviewer 3 Report

The manuscript is an interesting work related to a potential new therapy of hepatocellular carcinoma (HCC) by using TTFields in combination with a TKI (Sorafenib). However, the limited experimental design and paucity of strong data ask for more experiments to proof the feasibility of this combination for treating HCC

In vitro experiments sub-section in Methods sections lacks many experimental details (e.g., type of plate/flask, plating overnight or not before experiment, number of plated cells). A very important missing is not showing the actual number of rats included in the final analyses (the ones who successfully received therapy for more than 18 hours/day). This must be added.

What is puzzling in this investigation is showing functional data using two human cell lines and in vivo data using rat cells. While I do not know if the authors have the technology to perform TTFileds in mice, where for sure they should have done xenograft models with the two human cell lines, why the rat cell line was not studied in vitro using the same experimental strategies as for human cells. This must be done and included for a better understanding of TTFieldds activity from in vitro to in vivo data. What are the p53 status and the apoptosis signaling pathway function in N1S1 cells? Maybe the cytotoxicity data will reveal a different better frequency.

Another puzzling experiment is the schedule for the in vivo work. Since the authors missed to add Figure 4A for timeline, based on Methods section the rats were treated for 5 days with TTFields and or sorafenib and a day later the rays were sacrificed. While a short “acute” follow-up is welcome the most important experiment should allow the follow-up for much more days to indeed observe the effect of TTFields added to sorafenib. I could not find an explanation for not including a long term follow-up. In my opinion, this is a key therapeutically experiment which must be done and included in the study.

There is no explanation why the most effective dose was 150 kHz and higher does actually decreases the killing. Can this dose observed in vitro on only tumor cells be translated to in vivo work where the tumor microenvironment is total different?

Why the in vitro experiment was performed for 72 hours and in in vivo for 120 hours?

Any explanation for why not using cloroquine in vivo to integrate better the in vitro data.

The authors claim that cytotoxicity was measured “by cell counting using iCyt EC800 (Sony Biotechnology) 123 flow cytometer, and expressed as a percentage relative to the control.” Does this imply that they counted the live cells and plotted the final data as percentage of untreated controls (as figure 1 suggests). However, is this a real cytotoxicity or a cell growth inhibition? Did they measure the adherent cells after trypsinization. Were the cells from supernatant counted (where are probably the majority of dead cells)? There is a big difference between a therapy which kills vs a therapy which induces a cellular arrest.

Since sorafenib acts also on angiogenesis, did the authors investigate if TTFields may interfere with anti-angiogenic effect sorafenib-mediated? Also, is there any evidence that TTF may prevent the pretty common resistance to sorafenib observed in clinic? Moreover, there is a discrepancy between the fold changes in tumor weight vs. volume in the combination group vs. untreated group. Did the authors check changes in blood vessels density. Were the mice perfused before collecting the tumors?

Was Mycoplasma testing done routinely? Were the cells checked also for authenticity?

Is there any explanation why the combination of TTFields and sorafenib did not induce a significant level of autophagy as compared to untreated animals which invalid the initial hypothesis that “concomitant application of sorafenib and TTFields may increase stress levels enough to tilt autophagy towards the cell death pathway”.

The statement “TTFields concomitant with sorafenib induced a significant increase in apoptosis’ in the abstract section is overstated. When compared with sorafenib alone there is practically no difference. Moreover, TTFields failed to increases apoptosis when added to sorafenib and compared to sorafenib alone in one out of two human cells line investigated

Finally, adding to all the above questions, I found a very weak Discussion section which must be extended. Moreover, in the Discussion section, the authors concluded that “TTFields display efficacy for treatment of HCC in vitro and in vivo, with an optimal frequency of 150 kHz”. This is not a correct statement. While for in vitro data, the authors have data, for in vivo they used only one frequency of 150 Hz. At least one different dose should have been studied for comparison since this is a completely different tumor environment than the in vitro one.

Reviewer 4 Report

In general, the manuscript is well written and provides novel and interesting data illustrating the possibility to enhance the efficacy of sorafenib in the therapy of hepatocellular carcinoma (HCC). This might be achieved by using the combination of so-called Tumor Treating fields (TTFields) with targeted drug, 

I have the following suggestions about this manuscript:

1) The authors demonstrate the efficacy of TTFields in vivo even when used as monotherapy. As shown in the Figure 4 C and D, TTFields were found less effective in terms of reducing the tumors volume and weight  when compared with sorafenib. However, no differences in expression of LC3  marker  were observed  between these groups (treated with TTFields or with sorafenib)(as shown in Figure 4D). Similarly, low evidence of apoptosis (expression of cleaved PARP) was found in these groups, as shown in Figure 4F. What is the mechanism illustrating higher efficiency of sorafenib against HCC?

2) Despite the expression of cleaved PARP was very low in the tumors treated with TTFields or sorafenib alone ( as shown in IHC-images in Figure 4F), the authors declare about ~ 20% of positive cells, as show in the graphs below IHC-staining. Similar, the graphs illustrating the LC3 expression are not in a proper fit with the images shown in Figure 4D.

 3) It will be much better to provide the data to explain the mechanisms illustrating why the monotherapy of TTFields or sorafenin induced autophagy, whereas the tumors treated with combination  developed the substantial apoptotic death of tumor cells.  

4) Since Annexin V/7-ADD data was not convincing and the authors observed the minor increase of apoptotic cells after HCC cells were treated with combination of TTFields and sorafenib ( when compared to the cells treated with TTFiealds and sorafenib alone), I suggest to run the WBs to examine the expression of  the cleaved forms of PARP and caspase-3 ( for both HCC cell lines). This might be helpful and make the in vitro data more relavant with the data shown in vivo.  

Minor:

1) Figure 4A is missing. 

2) the different HCC cell lines were used for in vitro and in vivo experiments, therfore making difficult to compare these data.  

Round 2

Reviewer 3 Report

I thank the authors for their extensive work and comprehensive review of the manuscript. The revised manuscript is trying to be responsive to previous review comments. However, the new information added, instead to shed light, it brings more confusion.

Overall the author’s main conclusion in the abstract is overstated and is not supported by many questionable data and experimental design.

As I see the results in the actual setting, adding TTFields treatment demonstrated a toxic effect when added to sorafenib.

While I totally understand that the main reason for sacrificing the rats one day after last TTF, as authors mentioned in the letter “it was non-ethical to continue the study further”, this should have applied only for the rats belonging to control group. Majority of rodent study in cancer research lose the control group early but the treated groups are followed up for much longer to delineate indeed the efficacy vs. toxicity. It is practically impossible to appreciate a therapeutic effect in such a short time and upon monitoring effect for 1 day. Kaplan-Meyer survival curves are mandatory when toxic treatments are combined and compared. Is this indeed an anti-tumor effect or a very toxic effect to the liver (including tumor). Were there any investigation performed in animals to demonstrate a local/systemic toxicity? Were normal liver areas investigated? Were the IHC slides blinded to the pathologist(s)?

The in vivo model appears to be performed only time which question the validity of data. I am not sure why the randomization of rats was unequally, practically 50% in TTF groups vs. control and sorafenib alone. The rigor science recommends at least 2 independent experiments with at least 7 animals randomized per group.

The new western blot data are puzzling me. First of all I do not understand why it is so much variability between GADPH signals in all blots in the paper, especially within the same experiment. The reference protein should have the same intensity since this is the control for equal protein loading. Moreover, many bands belonging to all investigated proteins are truncated, fractured and I identified a lot of troubleshooting in bands due the presence of bubbling when running the blots. When come to densitometry, and especially when we look at the small range of fold changes between groups and controls, these troubleshooting can make huge differences. As a rule of thumb, when a housekeeping protein gives problems in immunoblotting, there are many other classic protein to switch the investigation (b-actin, vinculin, Ku, etc.). Otherwise, all protein investigation appears as a consequence of random effect with some fishing expedition.

Another weird observation is the new figures 3D, E, and F. Despite showing the same fold changes graphs in letter to reviewer and updated manuscript, the bands are different and all over the place in the mentioned files. The bands for 48 hours figure 3D (right panel), 6 and 24 hours for figure 3E and 48 hours for figure F are completely different between the letter and mansucript. What is even more confusing is that in figure 4 for time points 6 and 24 hours, where, despite keeping the same GAPDH bands, the cleaved PARP data are changed. There is no way to analyze bands for a protein using the GAPDH data from another experiment. Which data should be believable? This way to present the data questions a lot the quality of experimental work and consequently and importantly the final analysis of data and the concluion claimed by the authors.

Were these western blots (Figure 3D-F) repeated 3 times as the material and method mentions?

As a final observation, the Conclusion section, the hallmark of any work, is weird and poorly written. It looks like a speculative discussion section with no clear point of view of the authors and general conclusions of their presented work.